# Microstructure Evolution and Mechanical Properties of As-Cast and As-Compressed ZM6 Magnesium Alloys during the Two-Stage Aging Treatment Process

**DOI:** 10.3390/ma14247760

**Published:** 2021-12-15

**Authors:** Jia Fu, Su Chen

**Affiliations:** 1School of Material of Science and Engineering, Xi’an Shiyou University, Xi’an 710065, China; su_chen_vip@163.com; 2School of Material of Science and Engineering, Taiyuan University of Science and Technology, Taiyuan 030024, China

**Keywords:** ZM6 magnesium alloy, heat treatment, microstructure, mechanical properties

## Abstract

In the present study, different solid solution and aging processes of as-cast and as-compressed ZM6 (Mg_2.6_Nd_0.4_Zn_0.4_Zr) alloy were designed, and the microstructure and precipitation strengthening mechanisms were discussed. After the pre-aging treatment, a large amount of G.P. zones formed in the α-Mg matrix over the course of the subsequent secondary G.P. prescription, where the fine and dispersed Mg_12_(Nd,Zn) phases were precipitated at the grain boundaries. The pre-aging and secondary aging processes resulted in the Mg_12_(Nd,Zn) phase becoming globular, preventing grain boundary sliding and decreasing grain boundary diffusion. Meanwhile, precipitation phase â″(Mg_3_Nd) demonstrated a coherent relationship with the α-Mg matrix after the pre-aging process, and after the secondary aging phase, Mg_12_Nd increases and became semi-coherent in the matrix. Compared to an as-cast ZM6 alloy, the yield strength of the as-compressed ZM6 alloy increased sharply due to an increase in the yield strength that was proportional to the particle spacing, where the dislocation bypassed the second phase particle. Compared to the single-stage aging process, the two-stage aging process greatly improved the mechanical properties of both the as-cast and as-compressed ZM6 alloys. The difference between the as-cast and as-compressed states is that an as-compressed ZM6 alloy with more dislocations and twins has more dispersed precipitates in the G.P. zones after secondary aging, meaning that it is greatly strengthened after the two-stage aging treatment process.

## 1. Introduction

The ZM6 alloy (Mg-Nd-Zn-Zr) is a typical magnesium alloy that is strengthened by solid solution and aging processes and that has high strength, a small microporosity tendency, excellent mechanical properties, and good casting properties. Thus, it is commonly used in the aerospace and satellite communication fields due to its obvious age hardening effect [1,2,3,4]. Rare earth elements (RE) have a large solid solubility limit in magnesium, and that solid solubility decreases sharply when the temperature decreases, resulting in a large amount of supersaturation being obtained. For magnesium alloys without Al, such as ML10, ZE41, WE43, ZK60, AM-SC1, MRI202S, and ZM6, the grain refinement element is adopted [2,3,4]. The common heat treatment process is a solution treatment with the addition of artificial aging. The Mg-RE-Zr series alloy can be used at temperatures above 200 °C. As the diffusion rate of RE is slow, a dispersion strengthening phase containing RE can be precipitated in the aging process, which also provides the alloy with high thermal stability and excellent creep resistance [5,6].

The ZM6 alloy is a kind of Mg-Nd-Zn-Zr alloy, and most of the intermetallic compounds that are formed in the RE–Mg binary system are magnesium–rich phases. For light rare earth metals, the REMg_2_ compound is the Laves phase that belongs to the MgCu_2_ cubic system. Meanwhile, for heavy rare earths, the Laves phase belongs to the MgZn_2_ hexagonal system [7,8]. The stability of these binary phases increases from light rare earth metals to heavy rare earth metals. The Mg-Gd-Nd-Zr and Mg-Gd-Y-Zr alloys have been studied, and the nucleation of the aging phase has been found to be promoted by introducing a large number of dislocations [9]. The precipitation sequence in Mg-Gd-RE is as following: SSSS (super-saturated solid solution)→â″(Mg_3_Gd, hcp, D0_19_)→â′(Mg_3_Gd, fcc)→â(Mg_5_Gd, fcc) [3]. When the Mg-Gd-Y-Zr alloy is at its peak-aged condition, aging at 200 °C for 10–16 h and fine â″ particles with a D0_19_ structure (*a* = 0.64 nm, *c* = 0.52 nm) are the dominant strengthening phase. Moreover, â′ precipitates with an fcc structure (*a* = 0.742 nm) are the dominant phase in the alloy following over-aging at 250 °C for 10 h [10]. Ohishi and Xu et al. [11,12] studied the transformation mechanism of the precipitated phases in Mg-Zn and Mg-Al-Zn alloys during graded aging. It was found that the distribution and morphology of the precipitated phases changed significantly after graded aging compared to after single aging, resulting in the mechanical properties being significantly improved. However, as the aging behavior of the ZM6 (Mg_2.6_Nd_0.4_Zn_0.4_Zr) alloy is more complex, there are few reports about the multi-stage aging treatment of magnesium alloys, and their multi-stage aging behavior has not been revealed yet. Therefore, the present study observes the precipitation behavior during the two-stage aging treatment process and the effect of the aging temperature and aging time on the mechanical properties (yield strength, tensile strength and elongation) of both as-cast and as-compressed ZM6 alloys by adjusting and controlling the pre-aging stage during the two-stage aging process. In this paper, the phase composition, microstructure evolution, and age-strengthening mechanism of ZM6 alloys were investigated by means of XRD, DTA, and TEM. The morphology, size and distribution of the precipitation phases were controlled by the TSAT process, and the corresponding mechanical properties were tested through a comparison of those properties under various treatments in order to obtain excellent properties and to provide optimized process parameters for performance control in practical applications.

## 2. Materials and Methods

### 2.1. Sample Preparation and Initial Properties of As-Cast and As-Compressed ZM6

#### 2.1.1. Casting Process, Chemical Composition, and Hot Compression

The ZM6 alloy was prepared using pure magnesium (Mg ≥ 99.9 wt%), the Mg-25.0 wt% Nd master alloy, the Mg-33.3 wt% Zr master alloy, and pure zinc (Zn ≥ 99.9 wt%). The addition of aluminum, iron, silicon, manganese, and magnesium oxide should be avoided because these elements and impurities hinder the grain refinement in zirconium. During the smelting process, the alloy was melted in a resistance furnace and was protected by CO_2_ and SF_6_ gas in order to ensure that the alloy liquid did not become oxidized, thereby reducing the loss of alloy elements by burning and improving the casting quality. The test alloy was added in the form of a mixed rare earth metal and was dried at 200 °C before casting, and it was then melted in a steam crucible furnace. The crucible and the other tools were cleaned and preheated with flux before use, and they were then cast in a metal mold. The ingot was preheated in a preheated crucible at 200 °C and kept at 720 °C for 25–30 min. After the alloy was completely melted, the pure Zn and Mg-Nd master alloys were added at 750 °C and were held there for5 min; then, the Mg-Zr master alloys were added at 780 °C and were kept there 5 min, and then the 1% refining agent was added and stirred. Finally, the master alloys were kept at 780 °C for 20 min, and the metal mould (160 mm × 130 mm × 20 mm) was poured at 720 °C.

The chemical composition of the as-cast ZM6 (Mg_2.6_Nd_0.4_Zn_0.4_Zr) alloy was within the standard content region as follows: 0.2–0.7% Zn, 2.0–3.0% Nd, 0.4–1.0% Zr, Si: 0.01%, Cu: 0.03%, Ni: 0.01%, Fe: 0.01%, others: 0.05% (all in mass fraction). The as-cast bar stock was then compressed at 400–450 °C and at a compression degree of 8%. The tensile strength and the elongation of the as-cast and as-compressed ZM6 heat-resistant magnesium alloys were tested by the vendor of the Xi’an Brake Branch of Avic Aircraft Co., Ltd. (Xi’an, China).

#### 2.1.2. Mechanical Properties of As-Cast and As-Compressed ZM6

The mechanical properties of the as-cast and as-compressed ZM6 are listed in Table 1.

From Table 1, the as-cast ZM6 showed a surface area reduction of 4.5%. Before the heat treatment, the tensile strength, yield strength, and elongation properties of the as-compressed specimens increased by 78.74 MPa, 31.62 MPa, and 7.83% compared to the as-cast properties. The mechanical properties of the as-cast and as-compressed ZM6 alloys were provided by the vendor of the Xi’an Brake Branch of Avic Aircraft Co., Ltd.

### 2.2. Experimental Heat Treatment Schedules and Property Measurements

#### 2.2.1. Heat Treatment Schedules of As-Cast and As-Compressed ZM6

The as-cast and as-compressed ZM6 were used to prepare specimens using the DK7740 wire cutting machine (Jiangsu Fangzheng CNC Machine Tool Co., Ltd., Taizhou, China), which was operated with a voltage of 5.0 V and a current of 2.2 A. Sample dimensions for the heat treatment and tensile test are shown in Figure 1.

The tensile sample had the flat section size that is seen in GB/T1177–2018 (the specifications are as in Figure 1a), and the tensile rate was 1 × 10^−3^ s^−1^. The pre-compressed cylindrical samples had a diameter of 10 ± 0.05 mm and a length of 10 ± 0.05 mm, as seen in Figure 1b.

The ZM6 samples were heated in the KF1600-I box resistance furnace (Tianjin Mafuer Technology Co., Ltd., Tianjin, China) with a heating rate of 7 °C/min. The protective agent that was used during the heat treatment process was iron sulfide (pyrite). Among processes, the T4 process was treated using a solid solution (525 °C × 4 h), and the T5 process was treated through aging (200 °C × 8 h) without the use of a solid solution. The T6 process was treated using a solid solution at 525 °C for 4.0 h and through aging at different aging temperatures and aging times. Both the as-cast and as-compressed samples were studied under four different conditions in order to find the effect of the aging temperature and the aging time on their mechanical properties. The heat treatment schedules of the as-cast and as-compressed ZM6 samples that were designed and carried out are listed in Table 2.

From Table 2, the specimens that were in both states were tested simultaneously, and meanwhile, the T6 single-stage aging treatment (T6_SSAT) samples and T6 with two-stage-aging treatment (T6_TSAT) samples were measured and discussed for comparison. For the TSAT process, the pre-aging process comprised heating at 200–300 °C with a holding time of 0.05–0.25 h, and the final-stage aging was at 200 °C with a holding time of 8.25 h. The T6 heat treatment process is usually adopted, and the solid solution process that is usually chosen for experiments is 525 °C for 4 h with cooling at 80 °C or above, which can completely dissolve the second phase into the matrix to obtain a supersaturated solid solution. For the SSAT process, the aging temperature was 100–400 °C with temperature intervals of 100 °C that were obtained by air cooling. For the TSAT process, the pre-aging temperature was 200–300 °C with temperature intervals of 25 °C that were obtained air cooling, the pre-aging times were 0.05 h, 0.10 h, 0.15 h, 0.20 h, and 0.25 h, and then the secondary aging treatment (200 °C × 8 h) was carried out. The T6_SSAT process took place at 200 °C × 8.25 h, and the T6_TSAT process took place at 200–350 °C × 0.05–0.25 h and 200 °C × 8 h. When both the as-cast T6 and as-compressed T6 samples were aging at 200 °C for 0–64 h, the aging times were chosen to be 0 h, 4 h, 8 h, 12 h, 16 h, 20 h, 32 h, and 64 h.

#### 2.2.2. Microstructure Analysis and Mechanical Properties Measurement

After the solid solution and aging processes outlined in Table 2, each specimen was polished and etched with a specific etchant (10 mL acetic acid + 4.2 g picric acid + 10 mL H_2_O + 70 mL ethanol) or with a 4% nitric acid alcohol solution for about 3–5 s. The as-cast, as-compressed, and heat-treated specimens were inlaid with cold-setting epoxy resin. The grain size was measured using the ASTM E112–88 chord length method in polarized light mode with no less than 200 grains. The precipitate phases were observed and analyzed by means of an XRD-6000 X-ray diffractometer (Shimadzu Corporation, Tokyo, Japan), a VHX-600E optical microscope (OM) (KEYENCE Co., Ltd., Osaka, Japan), a JSM-7100F scanning electron microscope (SEM) (JEOL Ltd., Tokyo, Japan) with an Oxford AZtecX-Max20 energy spectrum probe (Oxford Instruments, Oxford, UK) for the energy dispersive spectroscopy (EDS) analysis and with a JEM-2100Plus transmission electron microscope (TEM) (JEOL Ltd., Tokyo, Japan) after ion thinning. Hardness was measured using the Vickers hardness tester (the load was 50 gf, the dwell time was about 15 s, the spacing between indentations was about 0.1 mm). Moreover, V-notched Charpy-notched specimens that were 5 mm × 10 mm × 55 mm (GB/T229–1994) in size were tested to measure the toughness. Each specimen was tested three times and then averaged. The hardness, elongation, and impact toughness were determined using the FM-700/SVDM4R microhardness tester (Future Tech Enterprise Inc., New York, USA), the FLFS–105 slow strain rate tensile tester (FULE Instrument Technology Co., Ltd., Shanghai, China), and the C64-305 MTS universal tester (MTS Systems (China) Co., Ltd., Eden Prairie, MN, USA), respectively. The liquidus and solidus temperatures of the Mg-Nd-Zn-Zr alloy during the cooling process were analyzed using a TGA/DSC 3 + differential thermal analyzer (Mettler Toledo, Zurich, Switzerland).

## 3. Results and Discussion

### 3.1. Analysis of the Alloy Phase Diagram and Initial Microstructures

The ZM6 alloy is a high-temperature magnesium alloy that is of a similar composition and that has similar properties to the ZE41A alloy and RZ5 alloys. The Zn content in the designed ZM6 alloy was low and completely dissolved in α-Mg matrix. The ZM6 that was used in this study was analyzed by Inductively Coupled Plasma (ICPS–1000 III) as being 2.58 Nd%, 0.36 Zn%, and 0.44 Zr% (dissolved Zr). 

#### 3.1.1. Alloy Phase Diagram Analysis of Mg-Nd-Zn-Zr Alloy

The representative two binary phase diagrams of the Zn-Mg, Nd-Mg [13], Zr-Mg [14], and Mg-Nd-Zn-Zr alloys are shown in Figure 2.

From Figure 2a, according to the Mg-Zn binary phase diagram, the maximum solubility of Zn in Mg is 6.18 wt% under equilibrium conditions. As the composition of the Zn element in the ZM6 (Mg_2.6_Nd_0.4_Zn_0.4_Zr) alloy was 4%, which is under the equilibrium solidification condition, all of the Zn was dissolved in the Mg matrix, and the solidification rate of Zn was much higher than that of equilibrium solidification, with a small Mg-Zn mesophase being present along the grain boundary [10]. As seen in Figure 2b, the formed precipitated-phase Mg_12_Nd contained a large number of Mg atoms with a high melting point and good thermal stability [10,13]. Generally, the melting points of rare earth elements are relatively high, reaching 798–1663 °C, and their diffusion in the Mg matrix is relatively slower [2,3]. The strength of the Mg-Nd-Zn-Zr alloy increased by 0.5–1 times, and the limiting temperature increased to 350 °C compared to the AZ31 alloy [10]. Nd can improve the creep strength of the ZM6 alloy [10]. Figure 2c shows the Mg-Zr phase diagram and demonstrates that the peritectic reaction took place at 649 °C, when the Zr content was 0.58. It should be noted that the Zr content at the peritectic reaction point varied from 0.50 to 0.76 in different alloys, demonstrating an obvious grain refinement effect [14]. However, this diagram cannot explain the alloy that had a Zr content of less than 0.58%. The Zr addition amount should be 3–5 times that of the dissolved Zr, that is, the amount of Zr that is added in the ZM6 alloy should be 1.32–2.2% during the casting process. As seen in Figure 2d, the ZM6 alloy phase diagram is composed of the liquid zone, the Mg matrix, α-Zr, Mg_41_RE_5_, and Mg_12_RE. For the ZM6 alloy, the composition of the magnesium-rich compound REMg_12_ for the light rare earth metals and RE_5_Mg_24_ for the trivalent heavy rare earth metals formed a eutectic equilibrium with Mg. The Mg_12_RE phase exists below 234.6 °C. The pouring temperature should be above the liquid point of 549.5 °C.

In order to reduce the hot cracking tendency that Mg-Zn-Zr alloys have, the rare earth metal Nd was added into the ZM6 alloy, which dissolved in the Mg matrix. At the magnesium-rich end, the peritectic reaction and the solid-phase transformation occurs more easily, so a metastable alloy can be easily formed at this end. As the melting points of the rare earth metals increase, the phase diagram shows a smooth change, and several rare earth compounds gradually disappear. The addition of Nd (2.0–3.0%) greatly reduces the solid solubility of zinc in magnesium, and the eutectic that is distributed in the grain boundary is brittle. The higher the content of rare earth elements in the alloy is, the better the casting properties are, but the mechanicals properties will be lower.

#### 3.1.2. DTA Thermal Analysis Curve and Characteristic Temperatures

In the eutectic equilibrium state, the eutectic composition of the compound changes gradually along with the atomic number of the RE element, and the solubility of the RE in the solid Mg decreases gradually as the atomic radius of the RE element increases. The first derivative of the cooling curve is *dT_c_/dt*, which is related to the solidification reactions for the different phases, which can be determined to determine the critical solidification characteristics of as-cast ZM6 alloys. The heating temperature was 780 °C, and the heating speed was 40 °C/min. Figure 3 shows the differential thermal analysis (DTA) curves of the as-cast and as-compressed ZM6 alloys.

From Figure 3a, when the temperature reaches at 636.54 °C, the elements in as-cast ZM6 alloy are all dissolved, and the eutectic transition occurs between 567.91 °C and 586.90 °C. It can be observed from Figure 3b that two defined peaks are present in the as-compressed ZM6 alloy. The ZM6 alloy consists of dissolved Zn and RE in α-Mg matrix and a small eutectic Mg-Zn phase, Mg-RE phase, and Mg-Zn-RE phase. Due to the addition of RE elements, a certain number of eutectic compounds are formed. When the temperature is cooled to the primary α-Mg phase formation reaction at 644.56 °C (point A), the non-equilibrium eutectic reaction occurs at 560 °C [15] and ends at 527.89 °C, resulting in Mg_12_Nd compounds being obtained. The temperature of the whole eutectic reaction is within the range of 567.91 ± 10 °C.

### 3.2. Microstructure and Phase Analysis after Two-Stage Aging Processes

The microstructures and XRD comparisons for the as-cast, as-compressed (A.C.), and A.C. T6_SSAT ZM6 alloys are shown in Figure 4.

Figure 4a demonstrates that the as-cast microstructure is composed of two parts: a grey α matrix and a white island Mg_12_Nd phase that are distributed at the grain boundary. The β phase (Mg_12_Nd) exists at the grain boundary on its own, and the average grain size is about 40.5 µm. In Figure 4b, the grain is remarkably refined and is uniformly distributed after hot compression. The grain boundary phase is broken, dispersed, and then refined, with an average grain size of 25.6 µm, which is in contrast with the almost chrysanthemum-shaped grains that are seen in the as-cast ZM6 alloy. The cotton-wadding shape inside the grain is considered to be ZnZr_3_ or Zr-containing particles [16]. The small grains are mainly formed by slip and twinning, and the continuous dislocation movement along the slip surface promotes the nucleation of the aging phase. As seen in Figure 4c, the second phase disappeared at the grain boundary and was replaced by the dispersed cluster precipitates, most of which were distributed in the grain boundary. The eutectic phase was distributed on the grain boundaries in the form of islands, the shape of which depends on the growth of the primary α-Mg phase [2,10,13]. The grain boundary is shown to be clear and regular, and the strip phase near the grain boundaries no longer exists. From the XRD comparison analysis of the ZM6 alloys in Figure 4d, only the α-Mg matrix and Mg_12_Nd compounds can be detected, and the main diffraction peaks are all Mg diffraction peaks. It is evident from Figure 4d that the as-compressed specimen with the T6_SSAT process has more texture compared to the as-cast specimen, which has a marked effect on mechanical properties.

#### 3.2.1. SEM and TEM Analysis after Aging Treatments

Taking the two-step aging process as an example, the microstructures of the as-cast ZM6 alloy under various aging temperatures and aging times are shown in Figure 5.

Figure 5a–c are the microstructures after pre-aging at 250 °C, 275 °C, and 300 °C with the holding time of 0.20 h, respectively, which are composed of polycrystalline α-Mg and are unevenly distributed cluster precipitates. A spherical phase exists in the grain. From Figure 5a, the grain shape changes from a chrysanthemum-shaped to hexagonal structure, and the average grain size increases slightly, from 39.6 μm to 41.0 μm. Figure 5d–f correspond to microstructures that have been pre-aged at 275 °C for 0.15, 0.20, and 0.25 h and then re–aged at 200 °C for 8 h; the microstructures show cluster-like precipitates that are still inhomogeneous and that are dispersed in the α-Mg solid solution. It can be seen from Figure 4b that that Mg_12_Nd compounds are around the α-Mg grain boundaries and demonstrate a coarse network morphology with spicule and rod-shaped phases [17,18]. It can be seen that after the final aging step, the average grain size of the as-cast ZM 6 is 39.6–41.5 μm, and the average grain size increases slightly when the pre-aging time is prolonged. Moreover, the effect of the pre-aging temperature on the average grain size is greater than that of the pre-aging time.

The TEM and HRTEM of the as-compressed samples with SSAT at 250–300 °C for 8.25 h and TSAT at 275 °C for 0.15–0.25 h + 200 °C for 8 h were analyzed, as shown in Figure 6.

From Figure 6a, it can be seen that the β″ (Mg_3_Nd) phase is uniformly distributed in the α-Mg matrix, but its size is relatively uneven. From Figure 6b, the black flake β″ phase that is larger in size is completely coherent with the matrix, and its length is about 25 nm. Figure 6c shows cluster-like precipitates with uneven distribution. The precipitates are rod-shaped and are 100–1500 nm in length, and a thin-layered (about 400 nm) precipitation-free zone is formed near the grain boundary. The intersecting lamellae are dense point-like phases that are accumulated from the edges. As seen in Figure 6d, after pre-aging at 275 °C for 0.15 h, a large number of G.P. zones are formed in the α-Mg matrix. Figure 6c shows two typical dislocations, and a dislocation network can be observed as a characteristic subgrain boundary that is formed by the interaction of those dislocations. Figure 6d is a regularly arranged dislocation wall in which the same number of dislocations repel each other. Figure 6e shows the distribution of the β″ phase in the matrix after the 275 °C × 0.20 h + 200 °C × 8 h aging treatments. The distribution of the β″ phase in the α-Mg matrix is more dispersed than the one that formed as a result of treatment with the T6_SSAT process. Figure 6f demonstrates that the β″ phase has a length of about 10 nm and a thickness 2 nm after the TSAT process. Compared to the SSAT process, the β″ phase becomes smaller after the TSAT process.

#### 3.2.2. EDS and Phase Analysis after Aging Treatments

The SEM-EDS analysis of the as-compressed ZM6 after the aging process is shown in Figure 7. 

As seen in Figure 7a, the microstructure is composed of two parts: the gray portion is the α-Mg matrix, that is, the solid solution Zn, Nd, and other elements in the α-Mg matrix, and the grain is primary spherulite, which is present in a small amount in the eutectic phase and has an island-like distribution at the grain boundary [19]. The divorced eutectic α-Mg phase is dependent on the α-Mg growth that takes place during the primary phases. Meanwhile, the β phase (Mg_12_Nd) and the Zr-containing particles (Zn_2_Zr_3_) exist alone at the grain boundary [20]. From Figure 7b,c, after the pre-aging process, the precipitation phase has a coherent relationship with the matrix, and then the secondary aging phase increases and becomes semi-coherent with the matrix. Figure 7e shows that the as-compressed ZM6 is composed of the α-Mg matrix and Mg_12_Nd phase. In XRD pattern in magnesium alloys, the α-mg Matrix has a dense hexagonal structure, with the precipitation of different rare earth phases and alloy phases, thus the matrix of the magnesium alloys can be strengthened [21,22,23,24,25]. From Figure 7f, the peak that is obtained by the TSAT process is higher than that that was obtained by the SSAT process, and the time that was needed to reach the peak was shorter due to the fact that there was a higher density of G.P. zones and a higher dispersion after the TSSAT process.

The EDS analysis in Figure 7 after the SSAT and TSAT processes is listed in Table 3.

As per the ESD analysis in Table 3, the second phase under the T6 condition is mainly a fine and dispersed â″ phase [26,27] when the peak hardness of the aging state is reached. Zr is used as a grain refiner in the ZM6 alloy, and the elements Zn, Nd, Zr, and Mg are present at the spot positions of the smaller particles.

In order to further analyze the phase precipitates, the TEM, XRD, and EDS for the phase analysis before and after the two-stage aging process are shown in Figure 8.

Figure 7a shows the TEM image and EDS energy spectrum of the grain boundary precipitates. The precipitates are combined with the matrix, and the precipitates are massive, ranging from about 200–300 nm in size, and the grain size varies greatly, with some subcrystals appearing. In addition to the second phase at the grain boundary, a small number of lamellar and granular phases were found in the inner (IC) and grain boundary precipitates (GBP). Most of the lamellar phase is suspended in the crystal. There are two types of granular phases that are present in Figure 7a: one is a disorderly distribution in and around the grain boundary, where the size is less than 200 nm, and the other is a directional arrangement along or near the grain boundary in the G.P. zones and is needle-shaped. The grain boundary precipitate phase exists between the intracrystalline precipitates during the two-stage aging process. A flaky phase, spotted-state phase, and half-elliptical phase also exist. The pre-deformed specimen consists of not only the β″/β′ precipitates that are distributed, it also includes the globular or rod-shaped β_1_ (FCC) precipitates on the dislocations in the matrix [8]. From Figure 7b, after pre-aging appears, the G.P. zones are totally coherent with the matrix, and these G.P. zones present as a heterogeneous core demonstrated limited distribution of the precipitated phase after the secondary aging process. It is generally believed that α-Zr and Mg have the same h.c.p lattice structure. The Zn_2_Zr_3_ phase that precipitated in the solid solution process did not undergo any obvious changes during the aging process [20]. The EDS results show that the atomic ratios of Mg, Zn, and Zr are 54.18%, 18.51%, and 27.31%, respectively.

Figure 7c shows the semi-elliptical precipitates that are found in the grain boundaries. The energy spectra of the flaky phase, granular phase, and semi-elliptic precipitated phase are similar and are demonstrated to be composed of Mg and Nd and to belong to different forms of the same substance. The precipitated Mg_12_Nd phase in the as-compressed microstructure has three forms: the bulk phase in the grain boundary, the flake phase in the grain boundary, and the granular phase near the grain boundary. Figure 7c shows the intragranular electron diffraction pattern of the selected area. The bright large spots in the calibration diagram are the diffraction patterns of the α-Mg matrix [21,22,23,24,25], while the small spots in the 1/2 matrix are those of Mg_12_Nd and demonstrate somewhat of a coherent relationship with the α-Mg matrix. Figure 7d shows the precipitation phase of Mg_12_Nd at the grain boundary. There are obvious precipitates at the grain boundary, linear precipitates near the grain boundary, and spherical precipitates with a diameter of 1–2 μm that are distributed in the crystal. The precipitates that are at the grain boundary are in either the Mg_9_Nd or Mg_12_(Nd, Zn) phase [19,20]. These particles are supposed to be Mg_12_Nd, and the smaller particles are arranged in a linear fashion around dislocations or grain boundaries [28,29].

#### 3.2.3. Precipitation Mechanism and Strengthening Mechanism

After the two-stage aging process, the tensile surface fracture at ambient temperature was determined and is shown in Figure 9.

Figure 9a shows that the as-cast ZM6 alloy has more obvious ductile fracture characteristics with an obvious tearing line and with spherical particles in the dimples. Moreover, the tensile deformation mechanism of the precipitation phase is mainly conducted by slip and twinning, and the dislocations are shown to demonstrate a continuous flow along the slip surface. Figure 9a shows how cleavage fractures and torn edges appeared; the intergranular fracture is the main mode and is partly along the mixed transgranular. In Figure 9b, the macroscopic fracture of the as-compressed ZM6 alloy is flush, and the granular fracture is depicted with a darker color. As the deformation of the magnesium alloy is not coordinated, and the crack source is easy to produce in the precipitation phase [30,31], which leads to a decrease in the elongation. In the durable fracture that is caused by aging in Figure 9b, there are many cracks in the grain boundary, and the grain boundary crack is more obvious. Although the tensile fracture surface is ductile, Mg(Zn)Zr compound particles still exist in the deep part of the dimples. In fact, the dislocation slip and deformation twins of wrought magnesium alloys often interact [32,33], which has an important effect on mechanical properties

The precipitation formation mechanism that takes place during the aging process is shown in Figure 10.

Figure 10a shows the schematic diagram of the precipitate transformation mechanism in a single-stage aging process. Generally, the existence of a G.P. zone ranges from room temperature to 180 °C, â″ ranges from 180 °C to 260 °C, and â′ ranges from 200 °C to 320 °C. Meanwhile, â exists at a higher temperature range, from 300 °C to the solvus temperature of about 550 °C [31,32]. The precipitation order is found to be [31,32,33]: supersaturated solution→G.P. zones→α(hcp)→β″(D0_19_)→β′(fcc)→β(bct), and the grainy precipitate is varied: the phaseflaky phase, spotted state phase, and half-elliptical phase. The G.P. zones are needle shaped and are located along the (000l) direction, as shown in Figure 10b. When a single-stage aging treatment is applied at 200 °C for 8 h, heterogeneous β″ phases [3,30] are rapidly precipitated from a supersaturated solid solution and grow rapidly due to the high aging temperature and due to early aging. However, in a certain range near the precipitates of the β″ phase in the early stage of aging, a dilution zone of solute atoms will be formed. As the aging time increases, the dilution zone will expand as the β″ phase continues to grow in the depleted zone of solute atom, making it difficult for the β″ phase to grow a nucleus during the later aging period. Therefore, after single-stage aging at 200 °C for 8.25 h, the size and dispersion of the β″ phase are larger and lower. Figure 10c shows the schematic diagram of the phase transformation and precipitation mechanism during the two-stage aging process. Compared to single-stage aging, the β″ phase density becomes larger, the size becomes smaller, and the β″ phase distribution is more dispersed [34,35,36]. During the pre-aging process, a large number of dispersed G.P. zones are first formed in the matrix. During the two-stage aging process, heterogeneous nuclei from the β″ phase in the G.P. zones increase and form a large number of dispersed β″ phases. As the heterogeneous nucleus of the â″ (Mg_3_Nd) phase and the â″ region increase, a large number of dispersed â (Mg_12_Nd) phases are formed [8]. According to references [17,18,19], the â″ and â′ phases are both plate-shaped and tend to be $\left\{11\bar{2}0\right\}$ and $\left\{10\bar{1}0\right\}$. The â′ phase (probably Mg_9_Nd or Mg_41_Nd_5_) demonstrates nucleates with a hexagonal structure on the dislocations [19] and demonstrate an apparent parallel orientation at ${\left(11\bar{2}0\right)}_{\beta \prime }∥{\left(10\bar{1}0\right)}_{Mg}$ and ${\left(\bar{1}014\right)}_{\beta \prime }∥{\left(0001\right)}_{Mg}$. The â″ (Mg_3_Nd, *a* = 0.64 nm, *c* = 0.52 nm) is perfectly coherent with the D0_19_ superlattice, that is, $a_{\beta \prime \prime }=a_{Mg}$, $c_{\beta \prime \prime }=c_{Mg}$, $a_{\beta \prime \prime }∥a_{Mg}$, and $c_{\beta \prime \prime }∥c_{Mg}$. The â (Mg_12_Nd) particles are incoherent and demonstrate a body-centred tetragonal symmetry of *a* = 1.031 nm, *c* = 0.593 nm [19], as shown in Figure 10d.

### 3.3. Experimental Tensile Properties under Various Heat Treatments

#### 3.3.1. Effect of Aging Temperature on Tensile Properties of As-Cast and As-Compressed ZM6

After the solid solution process (525 °C × 4 h), for the T6_SAST process, the aging temperature region was 100–400 °C, and the aging time was 8.25 h. Moreover, for the T6_TAST process, the pre-aging temperature region is between 225 °C and 300 °C, the pre-aging time is 12.0 min, and the final aging is at 200 °C for 8 h. The measured mechanical properties as well as the grain size under various aging temperatures are shown in Figure 11.

In Figure 11, the ultimate tensile stress (UTS) and yield stress are shown to decrease as the aging temperature increases. As the pre-aging temperature increases, the hardness at room temperature first increases and then decreases after the final aging process. When the aging time is extended, the overall strength during high-temperature tensile deformation gradually increases to its maximum value and then rapidly declines. For an aging process that takes place after the solution process, when the aging temperature increases, the microstructure demonstrates no significant difference, and the aging temperature has little effect on the grain size. The higher the aging temperature, the shorter the supersaturation of the solid solution and the shorter the time to reach the strength peak are. The lower the aging temperatures of the as-cast and as-compressed ZM6 alloys are, the higher the maximum hardness is. As the pre-aging temperature increases, the diffusion rate of alloying elements accelerates, and the phase is rapidly separated, so the required time is shorter. The mechanical properties of the both as-cast and as-compressed ZM6 alloys are very good at 200 °C, with an average relative fine grain size about 40 μm and 25 μm. In particular, there is a very good instantaneous tensile yield at 275 °C during the two-stage aging process. 

#### 3.3.2. Effect of Aging Time on Tensile Properties of As-Cast and As-Compressed ZM6

After the solid solution process (525 °C × 4 h), for the T6_SAST process, the aging temperature region was 275 °C, and the aging time was between 8 h and 20 h. Moreover, for the T6_TAST process, the regulation of the pre-aging temperature region was 275 °C, the pre-aging time was 6.0–15.0 min, and the final aging temperature was at 200 °C with a holding time of 8.0 h. The effects of the aging time on the mechanical properties that were affected by the aging process are shown in Figure 12.

After solution treatment, the length of time that was required for the subsequent aging process had an important effect on mechanical properties. From Figure 12a,b, the mechanical properties that were achieved by the T6_SSAT process reached their peak values after 16 h of aging, demonstrating a tensile strength of 230.32 MPa, a yield strength of 149.65 MPa, and hardness of 70.5 HV. For the T6_SSAT process, the highest tensile strength appears at the pre-aging time of 12 min, where σ_b_ = 294.46 MPa, σ_0.2_ = 188.72 MPa, and δ = 10.93%. When the aging time exceeds 16h, the tensile strength decreases, elongation increases, and over-aging occurs. During the aging process, the changes in the precipitation phase directly affect the changes that take place in the mechanical properties of the alloy. When the alloy is aged, the strength is increased, and the plasticity is decreased due to the precipitation of the â″ (Mg_3_Nd) and â (Mg_12_Nd) phases [37,38,39]. When the aging time is 16h, the UTS reaches 294.46MPa, and the elongation decreases to 10.93%. From Figure 12c,d, as the pre-aging time increases, the strength and hardness increase first and then decrease after the final aging period, and reach their maximum values at 275 °C, where the effect of pre-aging at 275 °C for 10 min is the best. For the two-stage aging process at 525 °C × 4 h + 275 °C × 12 min + 200 °C × 8 h, the comprehensive mechanical properties of the as-compressed ZM6 alloy are the best, demonstrating a yield strength of 188.72 MPa, a tensile strength of 294.46 MPa, and a hardness of 82.56 HV compared to the single-stage aging process. There is a very good short-term (12 min) aging limit between 250 °C and 325 °C.

The comparative mechanical properties of the as-compressed ZM6 alloy that were achieved by the T5 and T6_SAST processes are shown in Figure 13.

From Figure 13a, the strength increases gradually and then decreases slightly as the aging time increases. After solution treatment and aging for 16 h, the strength of the as-compressed ZM6 alloy reached its highest value of 268.84 MPa when the aging time was 16 h, and the elongation of the alloy also drops to the lowest point, but the elongation of 11.54% is still higher than that of the as-cast alloy. From Figure 13b, when the aging time was 16 h by T6_TSAT process, the elongation decreased from 11.95% to 10.62%. After aging for 16 h, during the longer aging time, the mechanical properties of the alloy did not change much. When the aging time for T6_TSAT the process is prolonged from 16 h to 64 h, the strength only changes by 3.1 MPa, and the elongation decreases by 0.5%. From Figure 13, the strength of the as-compressed ZM6 alloy by the T5 and T6_SAST processes increased along with the aging time, but the elongation of the alloy decreased gradually. The strength and hardness first increased and then decreased after the T5 treatment. During the whole aging process, the tensile strength and elongation changes showed opposite tendencies.

With the extension of the aging time, the precipitated phase at the boundary became thick and even. For the SSAT and TAST processes, when the dislocation bypassed the second phase particle, the yield stress was inversely proportional to the particle spacing, and the precipitation phase spacing was reduced.

#### 3.3.3. Comparison of Mechanical Properties under Various Heat Treatment Conditions

Comparison of mechanical properties as well as grain size in the as-cast and as-compressed alloys under the T5 and T6 conditions are shown in Figure 14.

Figure 14a shows the tensile strength at room temperature in different states. Figure 4b shows a comparison of the tensile properties between the as-cast state and the compression state. After the hot compression in Figure 14b, the strength is further increased because the microstructure is further refined and because the grain boundary per unit volume increases. During the tensile process, the non-complete microstructure homogeneity makes it difficult for the external load to be completely distributed and transferred to the neighboring grains, and these large grains become the source of the fracture, leading to the ductility decreasing. Under the same conditions, the mechanical properties of all of the compressed alloys are obviously better than those of the as–cast alloys. The reason for the differences in the mechanical properties is that more dispersed precipitates formed in the G.P. zones after secondary aging, thus strengthening the alloy [40,41]. Compared to the as-cast samples, the strength and elongation of the as-compressed ZM6 alloy increase. This is due to increases in both grain refinement and the grain boundaries, which can effectively prevent dislocation movement, resulting in increases in both the strength and elongation.

Mechanical properties, hardness, and the subsequent ZM6 heat-resistant magnesium alloy phases under different heat treatments are listed in Table 4.

As seen in Table 4, the ultimate tensile strength and the elongation of the as-cast ZM6 alloys are 155.36 MPa and 3.14%, respectively. After hot compression, the intermetallic compounds that broke down during the compression process also prevent the dislocation from moving, resulting in an increase in the tensile strength. For the as-compressed state, the σ_b_ increased to 294.46 MPa, and the elongation δ increased from 3.14% to 10.93% compared to that of as-cast ZM6. The matrix is strengthened by the appearance of a new grain boundary; thus, the strength and ductility are largely improved through hot compression. The ductility of the as-compressed alloys was higher than that of those alloys that were in the as-cast state. The UTS value of the as-compressed alloys obviously decreased, and the elongation increased after T4 treatment. After T5 treatment, the UTS value of the as-compressed alloy increased slightly, and the elongation the at break decreased. The elongation that took place during two-stage aging is lower than the elongation that took place during single-stage aging, where the elongation decreased by 5.12% and the tensile strength increased by 9.73%; thus, the heat treatment period is shortened. For the strengthening mechanisms, S.S.S. represents the solid solution strengthening, P.S. represents the precipitation strengthening, F.G.S. stands for the fine grain strengthening, and M.S. is the matrix strengthening. After the T4 solution aging treatment, a discontinued precipitated phase was produced at the grain boundary at the initial aging stage, and the elastic modulus of the matrix was therefore increased by solid solution strengthening (S.S.S.). After the TAST process, the precipitated-phase Mg_12_Nd with a BCT structure was linear or banded along the base plane of (0001) Mg, and the density near the grain boundary was higher than that in the crystal. The particles with linear distribution had an obvious strengthening effect on the Mg matrix. Moreover, there is a large stress field at the coherent interface between the matrix á-Mg and at the â phase, which hinders the dislocation movement. During the aging process, a large number of precipitated-phase Mg_12_Nd particles precipitated from a matrix. The smaller size and higher dispersion of the â″ phase were able to be obtained by decreasing the aging time within a certain temperature range [20], influencing the mechanical properties. The precipitation plays a pinning role for the dislocation and prevents the grain boundary from sliding, thus increasing the strength [42,43]. Therefore, after the two-stage aging treatment, the tensile strength is obviously improved, and there are slight changes in the elongation, which not only improves the mechanical properties but also reduces the heat treatment time.

In all, by studying the effect of the TSAT process on the microstructure and properties of ZM6 alloy, the aging time can be shortened effectively, allowing better properties to be obtained and better quality control of ZM6 products to be achieved.

## 4. Conclusions

The different a ZM6 (Mg_2.6_Nd_0.4_Zn_0.__4_Zr) aging processes were designed, and the phase precipitation and microstructure were analyzed after various heat treatments. Moreover, the mechanical properties that were achieved after various aging processes were discussed. The following results were obtained:(1)As-cast ZM6 alloys consist of α-Mg and Mg_12_Nd phases. The Mg_12_Nd phase is mainly distributed along the grain boundaries. Mg_12_Nd is distributed in the crystal and the grain boundary in the form of intersecting sheets, which are formed by the aggregation of the point-precipitated phases. The order of precipitation that occurs during the SSAT and TSAT processes was verified to be: supersaturated solution→G.P. zones→α(hcp)→β″(D0_19_)→β′(fcc)→β(bct).(2)As-compressed ZM6 alloys formed an α-Mg solid solution at 636.54 °C, and when it was cooled to 586.90 °C, it produced a Mg_12_Nd phase until the end of the eutectic reaction at 567.91 °C. There were no obvious differences in the microstructures of the as-cast and as-compressed ZM6 alloys; however, there was an obvious increase in the number of twins. Moreover, the twins that were inside and outside of the grain boundary were different in the as-compressed samples that did not undergo heat treatment.(3)After pre-aging at 275 °C for 12 min, a large number of small and dispersed G.P. zones were formed in the α-Mg matrix. After the two-stage aging treatment, the homogeneous lamellar Mg_3_Nd phase formed along the grain boundary, and the continuous precipitated fine Mg_12_Nd phase formed in the grains. During the second aging process at 200 °C for 8 h, the G.P. zone was used as the core of the β″ phase; thus, the size of the β″ phase was refined, and the density and dispersion of the β phase were increased at the end of the process.(4)When the pre-aging time increased, the elongation decreased slightly, and the tensile strength demonstrated an obvious increase. When the pre-aging process was at 275 °C for 12 min and when the secondary aging process was at 200 °C for 8 h, the tensile strength reached 294.46 MPa, which was 26.11 MPa higher than that obtained for the single-stage aging process at 200 °C for 8.25 h.(5)Compared to the T6 treatment, the tensile strength after the two-stage aging treatment improved great, showing values of up to 294.46 MPa. For the two-stage aging process, the best mechanical properties were obtained at 275 °C for 12 min + 200 °C for 8 h, where hardness values of 82.56 HV and tensile strength of 294.46 MPa were obtained compared to the values of 78.9 HV and 268.35 MPa that were achieved during the T6_SSAT process (aging at 200 °C for 8.25 h).

In brief, this basic investigation of the microstructure, phase precipitation, and mechanical properties after various aging treatments helped to reveal the strengthening mechanisms of ZM6 aircraft wheel work pieces. The optimized deformation parameters and crack propagation mechanism will be further investigated in the future.

## Figures and Tables

**Figure 1 materials-14-07760-f001:**
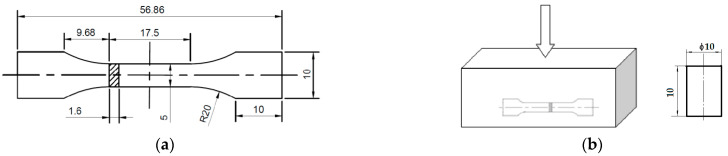
Sample dimensions: (**a**) Sample dimensions for tensile test; (**b**) pre-compressed deformation for tensile test and sample dimensions for solid solution and aging process.

**Figure 2 materials-14-07760-f002:**
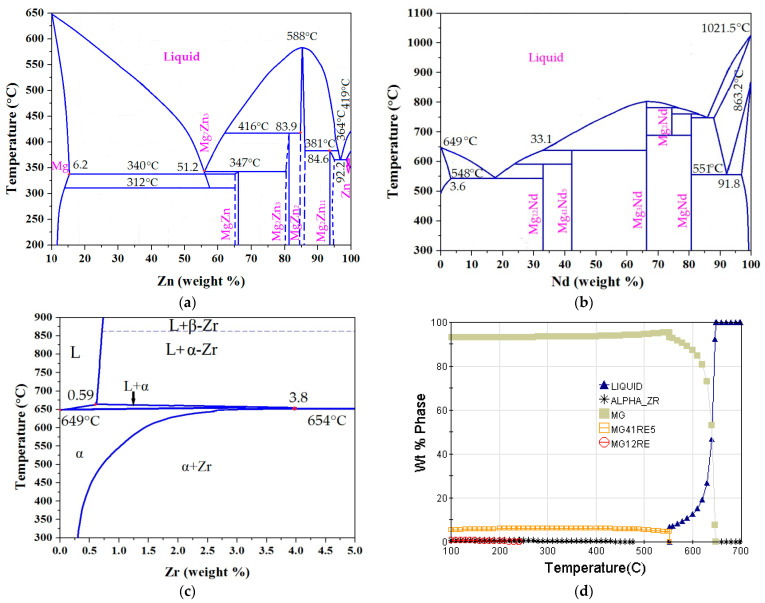
Solidification of binary and ternary phase diagrams: (**a**) Phase diagram of Mg-Zn alloy; (**b**) phase diagram of Mg-Nd alloy; (**c**) phase diagram of Mg-Zr alloy; (**d**) phase diagram of ZM6 (Mg_2.6_Nd_0.4_Zn_0.4_Zr) alloy.

**Figure 3 materials-14-07760-f003:**
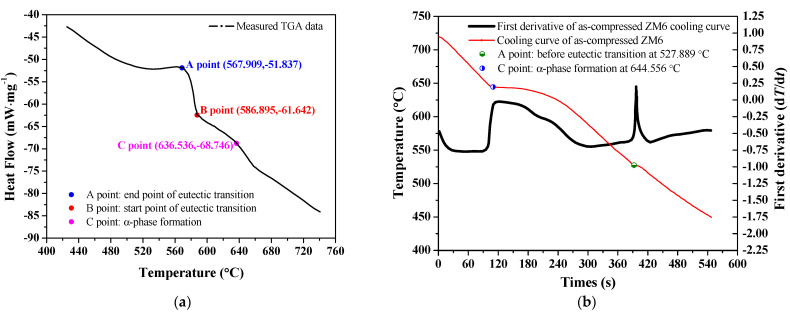
DTA curve: (**a**) DTA curve of the as-cast ZM6 alloy; (**b**) DTA curve of the as-compressed ZM6 alloy.

**Figure 4 materials-14-07760-f004:**
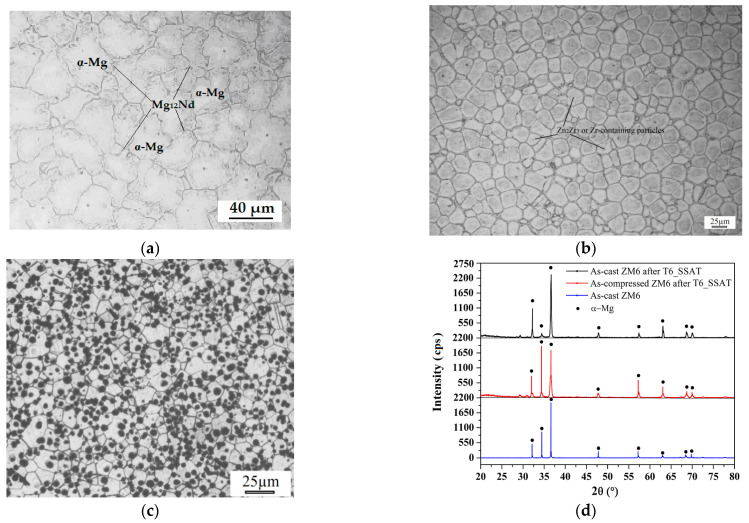
OM and XRD analysis using different heat treatment methods: (**a**) As-cast ZM6 alloy; (**b**) as-compressed ZM6 alloy; (**c**) as-compressed ZM6 alloy with T6_SSAT process; (**d**) XRD comparison of ZM6 alloys with T6_SSAT process.

**Figure 5 materials-14-07760-f005:**
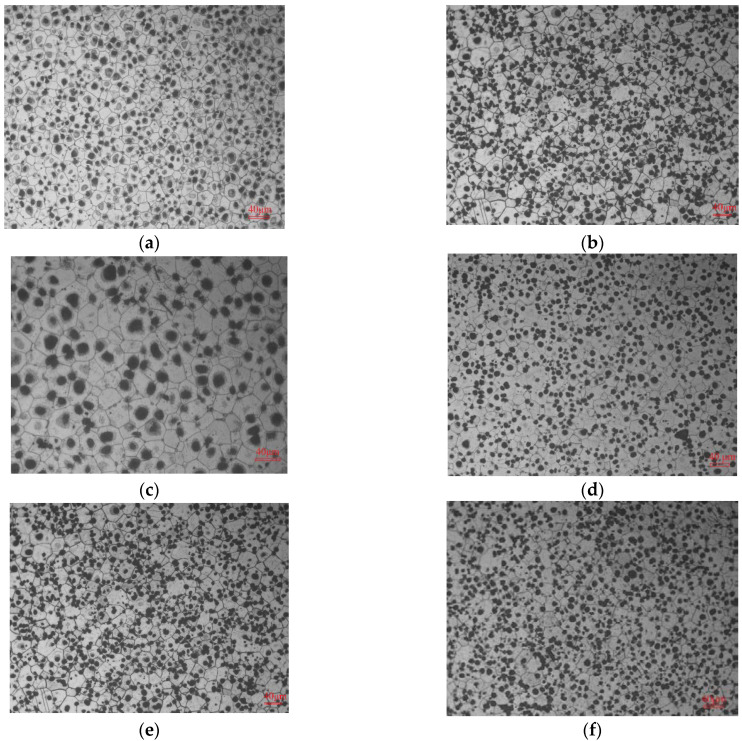
Microstructure of as-cast ZM6 alloy with different pre-aging temperatures and pre-aging times: (**a**) 250 °C × 0.20 h + 200 °C × 8 h; (**b**) 275 °C × 0.20 h + 200 °C × 8 h; (**c**) 300 °C × 0.20 h + 200 °C × 8 h; (**d**) 275 °C × 0.15 h + 200 °C × 8 h; (**e**) 275 °C × 0.20 h + 200 °C × 8 h; and (**f**) 275 °C × 0.25 h + 200 °C × 8 h.

**Figure 6 materials-14-07760-f006:**
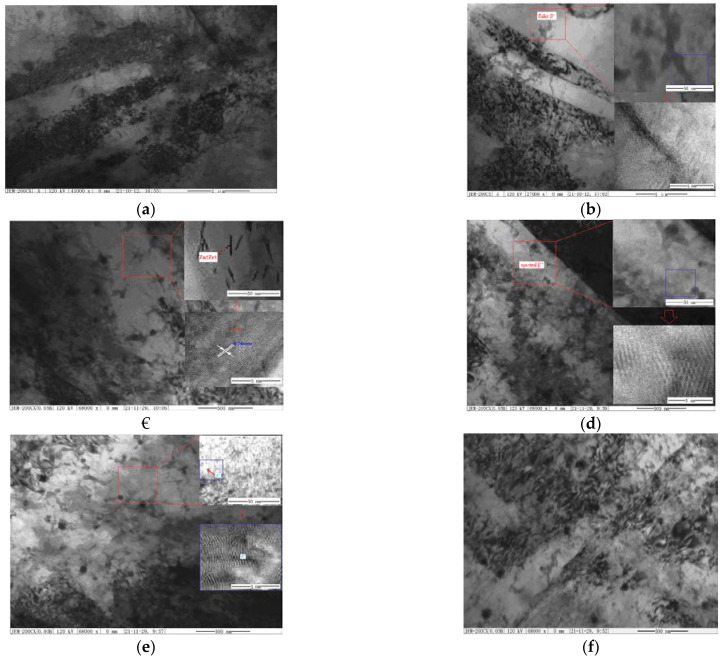
TEM images of as-compressed ZM6 alloy with single or two-step aging: (**a**) 250 °C × 8.25 h; (**b**) 275 °C × 8.25 h; (**c**) 300 °C × 8.25 h; (**d**) 275 °C × 0.15 h + 200 °C × 8 h; (**e**) 275 °C × 0.20 h + 200 °C × 8 h; (**f**) 300 °C × 0.25 h + 200 °C × 8 h.

**Figure 7 materials-14-07760-f007:**
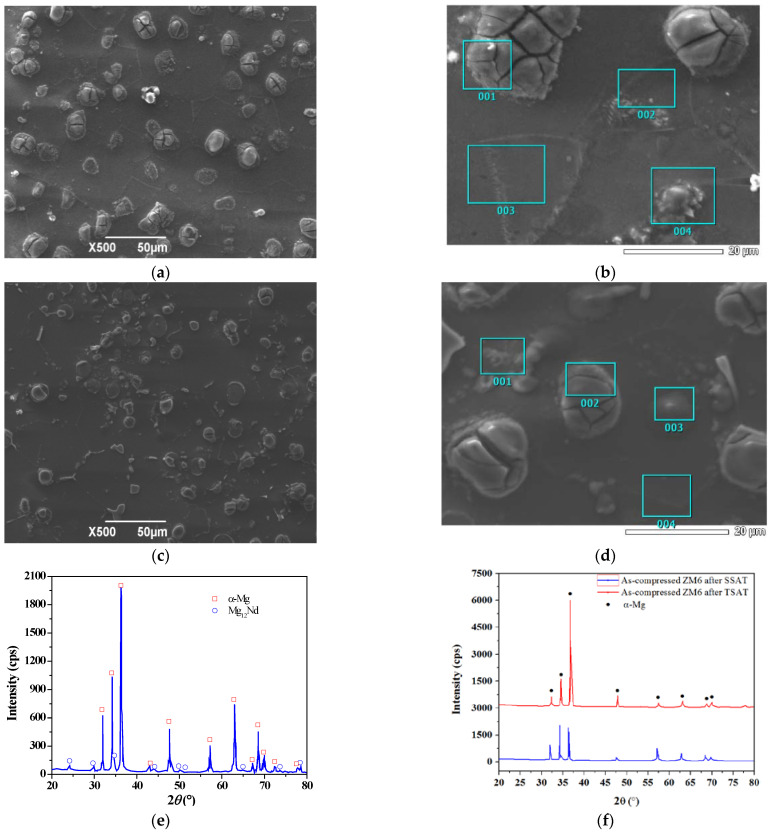
SEM-EDS and XRD analysis of as-compressed ZM6 alloy in discontinuous precipitated phases after aging process: (**a**) microstructure after the SSAT process; (**b**) EDS of sample after the SSAT process; (**c**) microstructure after TSAT process; (**d**) EDS of sample after the TSAT process; (**e**) XRD of sample after the TAST process; (**f**) XRD of sample after the SSAT and TSAT processes.

**Figure 8 materials-14-07760-f008:**
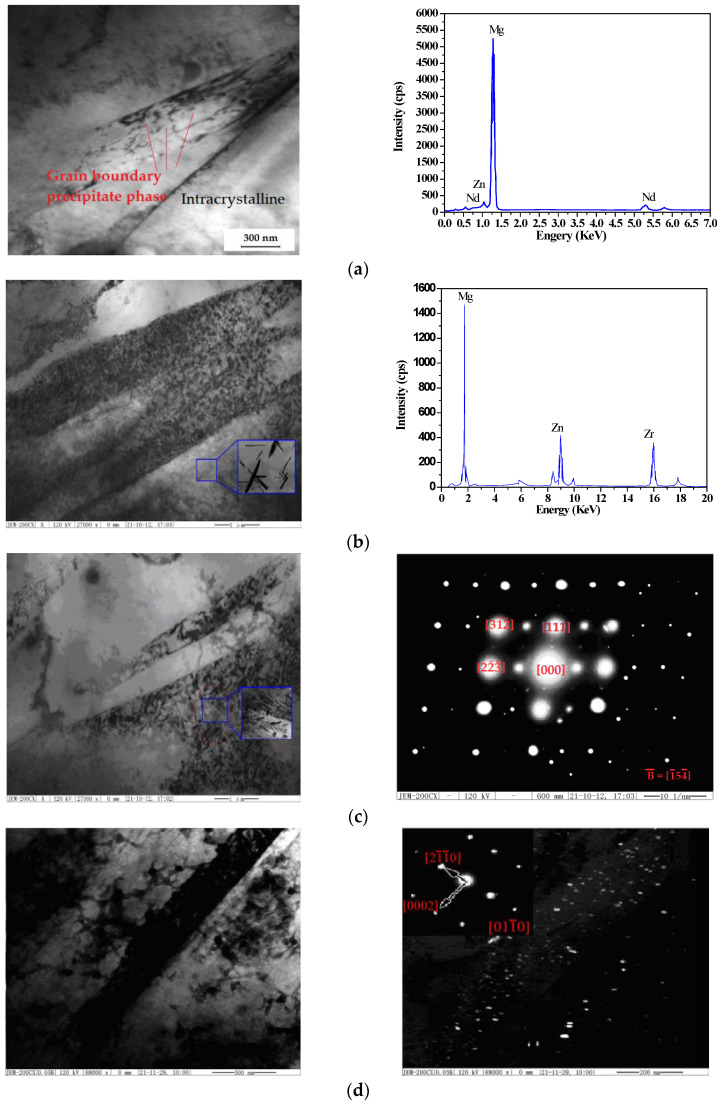
Phase analysis of as-compressed ZM6 before and after the two-stage aging process: (**a**) Grain boundary precipitate and EDS analysis after the TAST process; (**b**) TEM and EDS of continuously precipitated Zn_2_Zr_3_ phase after the TAST process; (**c**) TEM of the grainy precipitated phase and selected area diffraction patterns of spotted state precipitated Mg_12_Nd phase after the TAST process; (**d**) TEM bright-field image and TEM dark-field images of precipitates after the TAST process.

**Figure 9 materials-14-07760-f009:**
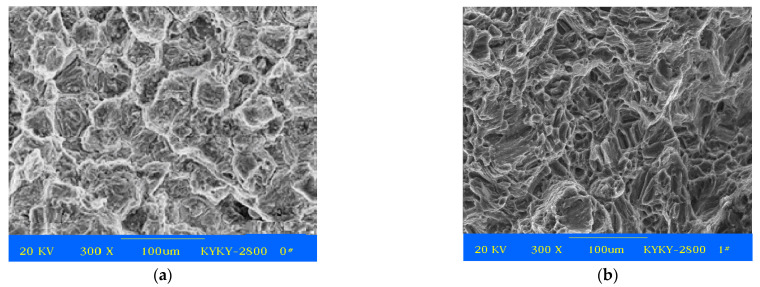
Tensile surface fracture of ZM6 after the two-stage aging process: (**a**) as-cast state; (**b**) as-compressed state.

**Figure 10 materials-14-07760-f010:**
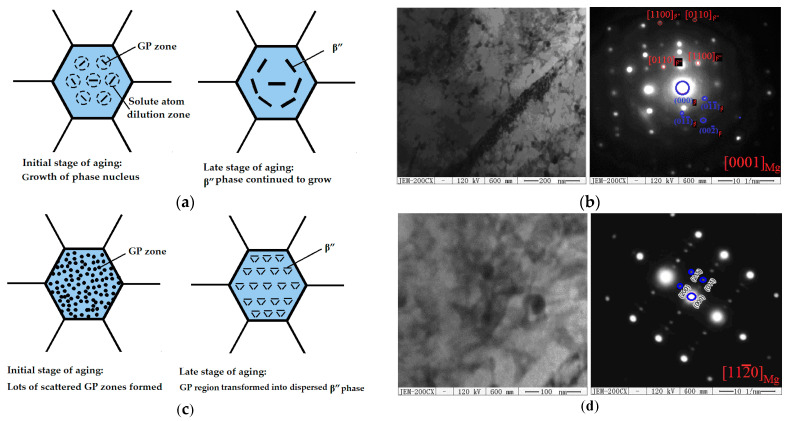
Precipitation formation mechanism by the SSAT and TSAT processes: (**a**) initial and final stages of the SSAT process; (**b**) TEM and SAED patterns of precipitates parallel to the (0001)_Mg_ zone axe by the SSAT process; (**c**) initial and final stages of pre-aging by the TSAT process; (**d**) TEM and SAED patterns parallel to (1120)_Mg_ by the TSAT process.

**Figure 11 materials-14-07760-f011:**
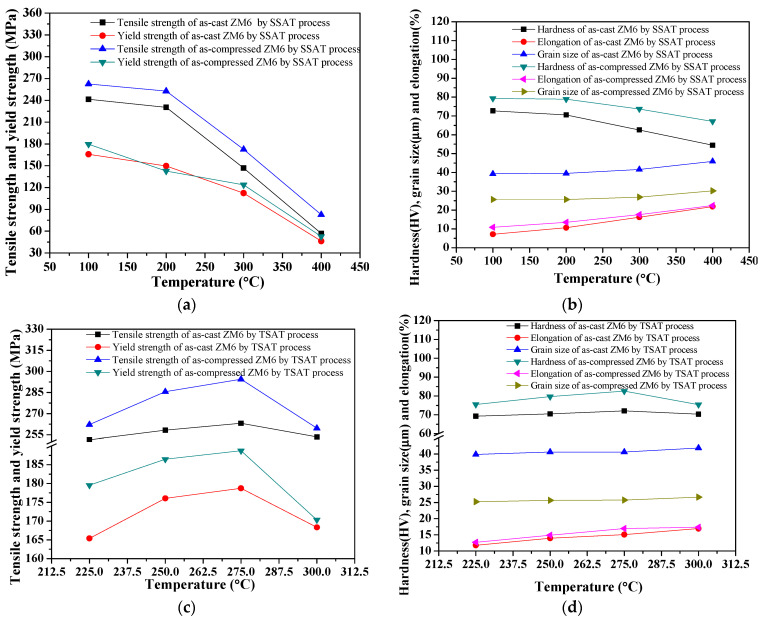
Mechanical properties and grain size of ZM6 alloys under various aging temperatures: (**a**) tensile and yield strength by SAST process; (**b**) hardness, elongation, and grain size by SAST process; (**c**) tensile and yield strength by TAST process; (**d**) hardness, elongation, and grain size by TAST process.

**Figure 12 materials-14-07760-f012:**
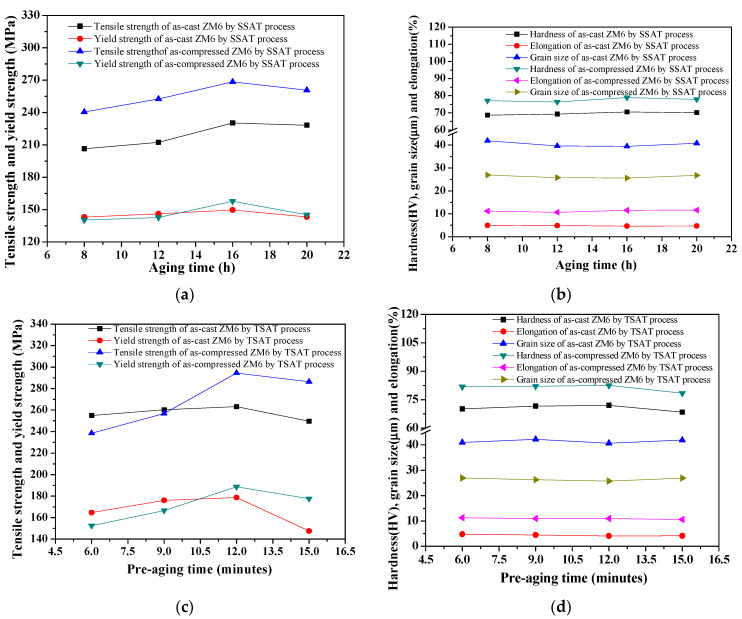
Mechanical properties and grain size of ZM6 alloys under various aging times: (**a**) tensile and yield strength by SAST process; (**b**) hardness, elongation, and grain size by SAST process; (**c**) tensile and yield strength by TAST process; (**d**) Hardness, elongation, and grain size by TAST process.

**Figure 13 materials-14-07760-f013:**
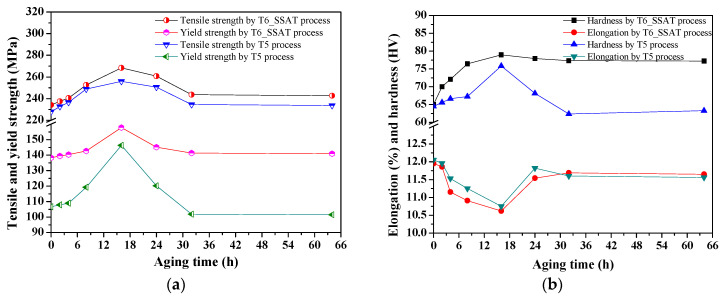
Comparative mechanical properties of as-compressed ZM6 alloy by T5 and T6_SAST processes: (**a**) tensile and yield strength; (**b**) elongation and hardness.

**Figure 14 materials-14-07760-f014:**
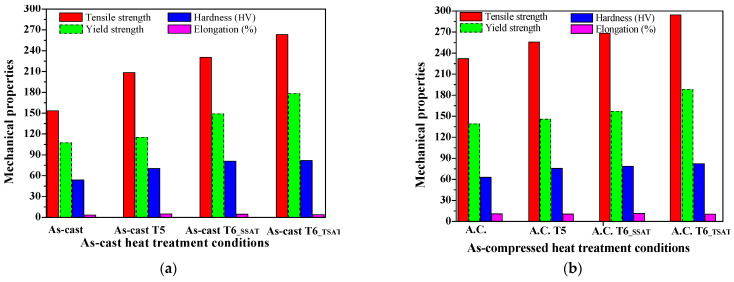
Mechanical properties of ZM6 alloys under different conditions: (**a**) mechanical properties under various as-cast treatment conditions; (**b**) mechanical properties under various as-compressed treatment conditions.

**Table 1 materials-14-07760-t001:** Tensile properties and hardness of as-cast and as-compressed ZM6 alloys.

	Tensile Properties				Hardness
	σ_b_ (MPa)	σ_0.2_ (MPa)	δ (%)	ø (%)	HV
As-cast ZM6	153.36	108.13	3.14	4.5	54.12
As-compressed ZM6	232.16	139.75	10.97	-	63.26

**Table 2 materials-14-07760-t002:** Schedules for the heat treatment experiments of the ZM6 alloys.

	Single-Stage Aging of ZM6		Two-Stage Aging of ZM6		
Condition	Solid Solution	Aging Process	Solid Solution	Pre-Aging Process	Final–Stage Aging Process
As-cast T5	-	200 °C × 8.25 h	-	275 °C × 0.25 h	200 °C × 8 h
As-cast T6	525 °C × 4 h	100–400 °C × 8.25 h	525 °C × 4 h	200–350 °C × 0.05–0.25 h	200 °C × 0–64 h
As-compressed T4	525 °C × 4 h	-	525 °C × 4 h	-	-
As-compressed T5	-	200 °C × 8.25 h	-	275 °C × 0.25 h	200 °C × 8 h
As-compressed T6	525 °C × 4 h	100–400 °C × 8.25 h	525 °C × 4 h	200–350 °C × 0.05–0.25 h	200 °C × 0–64 h

**Table 3 materials-14-07760-t003:** EDS spot analysis of ZM6 alloys after single-stage aging and two-stage aging processes.

Element	Spot Position	Mass %	Atom %	Spot Position	Mass %	Atom %	Spot Position	Mass %	Atom %	Spot Position	Mass %	Atom %
Mg K	Spot 1 in Figure 8a	60.46	88.26	Spot 3 in Figure 8a	83.72	96.64	Spot 1 in Figure 8b	75.75	93.71	Spot 3 in Figure 8b	81.70	95.70
Nd L		32.28	7.94		15.06	2.93		19.14	3.99		15.15	2.99
Zn K		6.34	3.44		0.47	0.20		4.74	2.18		2.64	1.15
Zr L		0.92	0.36		0.75	0.23		0.36	0.12		0.51	0.16
Mg K	Spot 2 in Figure 8a	78.24	94.27	Spot 4 in Figure 8a	79.65	94.82	Spot 2 in Figure 8b	60.67	88.15	Spot 4 in Figure 8b	85.98	96.98
Nd L		16.06	3.26		14.90	2.99		31.17	7.63		12.21	2.32
Zn K		5.05	2.26		3.68	1.63		6.95	3.75		1.31	0.55
Zr L		0.65	0.21		1.77	0.56		1.21	0.47		0.50	0.15

**Table 4 materials-14-07760-t004:** Comparison of tensile properties, hardness at room temperature, and strengthening mechanisms under various conditions.

Condition	σ_b_ (MPa)	σ_0.2_ (MPa)	δ (%)	Hardness (HV)	Strengthening Mechanisms
As-cast	153.36	108.13	3.14	54.12	M.S. + S.S.S.
As-cast T6-SSAT	230.32	149.65	4.63	70.50	M.S. + S.S.S. + P.S.
As-cast T6-TSAT	263.16	178.75	4.07	72.05	M.S. + S.S.S. + P.S.
As-compressed	232.16	139.75	10.67	63.26	M.S. + F.G.S.
As-compressed + T4	208.42	115.81	12.96	59.63	M.S. + S.S.S. + F.G.S.
As-compressed + T5	255.88	146.28	10.75	75.82	M.S. + F.G.S. + P.S.
As-compressed + T6_SSAT	268.35	157.65	11.52	78.92	M.S. + S.S.S. + F.G.S.+ P.S.
As-compressed + T6_TSAT	294.46	188.72	10.93	82.56	M.S. + S.S.S. + F.G.S.+ P.S.

## Data Availability

Data are available in a publicly accessible repository that does not issue DOIs. Publicly available datasets were analyzed in this study.

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
