# Peer review of "Microstructure Evolution and Mechanical Properties of As-Cast and As-Compressed ZM6 Magnesium Alloys during the Two-Stage Aging Treatment Process"

_materials, 2021, doi:10.3390/ma14247760_

Round 1

Reviewer 1 Report

Dear Authors,

You can find in the attached document several mistakes highlighted.

Also, your presentation of the samples used in the study is very confusing and hard to follow. I strongly recommend a more clear presentation of the samples used for the study, especially the treated ones.

My best regards.

Author Response

You can find in the attached document several mistakes highlighted.

Thank you for your highlighted mistakes of our original version. We've noticed some language issues suggested and we have also polished the content of the article, it's readable and available in the revised version. Also we reorganized the structure of this paper. All the repeated experiments and data this time were confirmed and verified by me and Chen Su. All English corrections are marked in red, seen in the revised version in details.

Also, your presentation of the samples used in the study is very confusing and hard to follow. I strongly recommend a more clear presentation of the samples used for the study, especially the treated ones.

Thank you, we have provided a clear presentation of sample used, which is as follows:

Reviewer 2 Report

The paper should be re-organized thoroughly to better present the results, figures, and figure captions. The amount of data and the number of pages should be reduced to briefly and concisely convey the idea of the paper. The figures should be re-arranged. Some plots have grain size, hardness, and elongation shown in the same plot, which is quite confusing. Please find a way to regroup them and present them better.

1.What is the main question addressed by the research? Is it relevant and interesting?

The paper contains a range of techniques used to investigate the ZM6 Mg alloys aiming to characterize their microstructure completely.  Furthermore, mechanical testing is correlated with the observed microstructure. Regarding the amount and diversity of data, the paper is acceptable for publication. However, in terms of their quality and how they are represented in the article, it has a lot of space for improvement. According to the report, the aim is to test the material under various thermal heat treatments to obtain the best mechanical properties, which is a general and commonly found description and is not related to answering any particular question.

2. How original is the topic? What does it add to the subject area compared with other published material?

The article compares single-step aging with two-step aging with various aging temperatures, holding time combinations, and as-cast with as-compresses ZM6 alloys. The investigation is quite widespread and not concentrated on one particular topic or comparison. 

3. Is the paper well written? Is the text clear and easy to read?
The paper has good grammatical English and it is easy to read but the text is not concise and it is quite confusing to convey the ideas.

4. Are the conclusions consistent with the evidence and arguments presented? Do they address the main question posed?

The conclusions related to mechanical testing are supported by the mechanical test results. However, the TEM imaging must have been of higher quality in order to draw conclusions about which phases are present in the magnesium matrix, as presented in this paper.  The TEM images lack the zone axis along which the observation was carried out and the main matrix direction of the Mg matrix. Moreover, there is no EDS analysis of the precipitates. To identify the phase of the precipitate, it is required shape, size, and orientation relationship with the parent matrix and/or EDS analysis

Author Response

The paper should be re-organized thoroughly to better present the results, figures, and figure captions. The amount of data and the number of pages should be reduced to briefly and concisely convey the idea of the paper. The figures should be re-arranged. Some plots have grain size, hardness, and elongation shown in the same plot, which is quite confusing. Please find a way to regroup them and present them better.

Thank your for your effective proposal. The structure of the paper is reorganized thoroughly according to your suggestions, which is as follows:

  1. Introduction
  2. Materials and Methods

2.1 Sample preparation and initial properties of as-cast and as-compressed ZM6

2.1.1 Casting process, chemical composition and hot compression

2.1.2 Mechanical properties of as-cast and as-compressed ZM6

2.2 Experimental heat treatment schedules and properties measurements

2.2.1 Schedules of heat treatments of as-cast and as-compressed ZM6

2.2.2 Microstructure analysis and mechanical properties measurement

  1. Results and discussion

3.1 Analysis of the alloy phase diagram and initial microstructures

3.1.1 Alloy phase diagram analysis of Mg-Nd-Zn-Zr Alloy

3.1.2 DTA thermal analysis curve and characteristic temperatures

3.2 Microstructure and phase analysis after two-stage aging processes

3.2.1 SEM and TEM analysis after aging treatments

3.2.2 EDS and phase analysis after aging treatments

3.2.3 Precipitation mechanism and strengthening mechanism

3.3 Experimental tensile properties under various heat treatments

3.3.1 Effect of aging temperature on tensile properties of as-cast and as-compressed ZM6

3.3.2 Effect of aging time on tensile properties of as-cast and as-compressed ZM6

3.3.3 Comparison of mechanical properties under various heat treatment conditions

  1. 4. Conclusions

1.What is the main question addressed by the research? Is it relevant and interesting?

The paper contains a range of techniques used to investigate the ZM6 Mg alloys aiming to characterize their microstructure completely.  Furthermore, mechanical testing is correlated with the observed microstructure. Regarding the amount and diversity of data, the paper is acceptable for publication. However, in terms of their quality and how they are represented in the article, it has a lot of space for improvement. According to the report, the aim is to test the material under various thermal heat treatments to obtain the best mechanical properties, which is a general and commonly found description and is not related to answering any particular question.

Thank you for your affirmation of the scientific research value of the mechanical properties after various heat treatments. In fact, we have expanded the analysis of this part, including the yield strength, the elongation, the reduction of section and the grain size after the quenching and tempering processes. We've noticed some language issues suggested and we have also polished the content of the article, it's readable and available in the revised version. Also we reorganized the structure of this paper. All the repeated experiments and data this time were confirmed and verified by me and Chen Su. All English corrections are marked in red, seen in the revised version in details. For your suggestion on the introduction provide sufficient background, we added the reference where the precipitation sequence in Mg-Gd-RE is given, which is very interesting and our research on the precipitation sequence is also investigated.

  1. How original is the topic? What does it add to the subject area compared with other published material?

The article compares single-step aging with two-step aging with various aging temperatures, holding time combinations, and as-cast with as-compresses ZM6 alloys. The investigation is quite widespread and not concentrated on one particular topic or comparison.

Thank you for your valuable comment. In the revised version, we focused on several research points:

  1. Although there is little difference in microstructure between as-cast and small-deformation pre-compressed state, the twin crystal is obvious and the average grain size is small in as-compressed samples.
  2. In comparison with single-stage aging, G. P. The more the region nucleates, the easier the formation of the fine Mg12Nd phase, so you get better mechanical properties.
  3. Particle shape is not the key to strengthening. Considering the distribution of the precipitates, the macroscopical direction of the precipitates distribution is parallel to the surface direction of (0001) Mg matrix after high temperature and short time aging, that is, the distribution of precipitates on the matrix surface is very dense. When as-compressed ZM6 alloys are deformed, the mechanism of deformation is mainly the matrix plane slip, and the dislocation moves along the parallel matrix plane. Therefore, when the precipitates act as an obstacle to the dislocation movement, the spacing between obstacles will be small due to its dense distribution on the matrix. Thus, the dense distribution on the matrix surface will increase the increment of critical shear stress, which can effectively increase the critical shear stress and make the strengthening effect remarkable.
  4. Is the paper well written? Is the text clear and easy to read?
    The paper has good grammatical English and it is easy to read but the text is not concise and it is quite confusing to convey the ideas.

Thank you for your suggestion, we rewrote the conclusion section to clarify the research focus and highlights of this paper, which is as follows:

  1. 4. Conclusions

The different aging processes of ZM6 (Mg2.6Nd0.4Zn0.4Zr) are designed, and the phase precipitation and microstructure after various heat treatments are analyzed. Besides, mechanical properties after various aging processes are discussed. Results are:

(1) As-cast ZM6 alloy consists of α-Mg and Mg12Nd phases. Mg12Nd phase is mainly distributed along grain boundaries. Mg12Nd is distributed in the crystal and the grain boundary in the form of intersecting sheets, which is formed by the aggregation of point precipitated phases. The precipitation order during the SSAT and TSAT processes is verified to be: supersaturated solution→G.P. zones→α(hcp)→β”(D019)→β’(fcc)→β(bct).

(2) As-compressed ZM6 alloy formed α-Mg solid solution at 636.54 °C, then cooled to 586.90 °C and produced Mg12Nd phase until the end of eutectic reaction at 567.91 °C. There is no obvious difference in microstructure of as-cast and as-compressed ZM6 alloys, except for the number of twins increased obviously. Besides, the twins is different inside and outside the grain boundary in as-compressed sample without heat treatment.

(3) After pre-aging at 275 °C for 12 min, a large number of small and dispersed G.P. zones were formed in α-Mg matrix. After two-stage aging treatment, the homogeneous lamellar Mg3Nd phase is formed along the grain boundary, and the continuous precipitated fine Mg12Nd phase is formed in the grains. During the second aging at 200 °C for 8 h, the G.P. zone was used as the core of β” phase, thus the size of β” phase was refined and the density and dispersion of β phase were finally increased.

(4) With the increase of pre-aging time, the elongation decreases slightly and the tensile strength increases obviously. When the pre-aging process is at 275 °C for 12 min and the secondary aging at 200 °C for 8h, the tensile strength reaches to 294.46 MPa, which is 26.11 MPa higher than that of single-stage aging at 200 °C for 8.25 h.

(5) Compared with the T6 treatment, the tensile strength after two-stage aging treatment is greatly improved, up to 294.46 MPa. For two-stage aging process, the best mechanical properties are obtained at 275 °C for 12 min + 200 °C for 8 h, with hardness 82.56 HV and tensile strength 294.46 MPa, compared with 78.9 HV and 268.35 MPa of T6_SSAT process (aging at 200 °C for 8.25 h).

In brief, this basic investigation of the microstructure, phase precipitation and mechanical properties after various aging treatments helps to reveal the strengthening mechanisms of ZM6 aircraft wheel work pieces. The optimized deformation parameters, crack propagation mechanism, will be fatherly investigated in future.

  1. Are the conclusions consistent with the evidence and arguments presented? Do they address the main question posed?

The conclusions related to mechanical testing are supported by the mechanical test results. However, the TEM imaging must have been of higher quality in order to draw conclusions about which phases are present in the magnesium matrix, as presented in this paper.  The TEM images lack the zone axis along which the observation was carried out and the main matrix direction of the Mg matrix. Moreover, there is no EDS analysis of the precipitates. To identify the phase of the precipitate, it is required shape, size, and orientation relationship with the parent matrix and/or EDS analysis.

Thank you very much. We have reorganized this article and added the SEM-EDS analysis to improve the quality of this paper, which is as follows:

Round 2

Reviewer 2 Report

The authors have made the necessary adjustment and the paper is acceptable for publication as it is.